# Molecular Dynamics Simulations Correlating Mechanical Property Changes of Alumina with Atomic Voids under Triaxial Tension Loading

**Junhao Chang** *,†, **Zengtao Chen** and **James D. Hogan**

Department of Mechanical Engineering, University of Alberta, Edmonton, AB T6G 1H9, Canada
* Correspondence: jchang7@ualberta.ca
† Current address: Department of Mechanical Engineering, University of Alberta, Edmonton, AB T6G 2R3, Canada.

**Abstract:** The functionalization of nanoporous ceramics for applications in healthcare and defence necessitates the study of the effects of geometric structures on their fundamental mechanical properties. However, there is a lack of research on their stiffness and fracture strength along diverse directions under multi-axial loading conditions, particularly with the existence of typical voids in the models. In this study, accurate atomic models and corresponding properties were meticulously selected and validated for further investigation. Comparisons were made between typical material geometric and elastic properties with measured results to ensure the reliability of the selected models. The mechanical behavior of nanoporous alumina under multiaxial stretching was explored through molecular dynamics simulations. The results indicated that the stiffness of nanoporous alumina ceramics under uniaxial tension was greater, while the fracture strength was lower compared to that under multiaxial loading. The fracture of nanoporous ceramics under multi-axial stretching, was mainly dominated by void and crack extension, atomic bond fracture, and cracking with different orientations. Furthermore, the effects of increasing strain rates on the void volume fraction were found to be similar across different initial radii. It was also found that the increasing tension loading rates had greater effects on decreasing the fracture strain. These findings provide additional insight into the fracture mechanisms of nanoporous ceramics under complex loading states, which can also contribute to the development of higher-scale models in the future.

**Keywords:** molecular dynamics; crack propagation; multiaxial loading; atomic voids

## 1. Introduction

The application of advanced ceramics, such as alumina ($Al_2O_3$), has been widespread in numerous engineering applications, including microelectronic devices [1], protection systems [2,3], and protective coatings [4,5]. Porous ceramics have gained significant importance in industry due to their high strength and lightweight characteristics [2,3]. At the nanoscale, the failure mechanism of porous ceramics is determined by factors such as the pore size, loading rates, and orientations. Hence, it is crucial to develop a thorough understanding of the fracture modes and establish adequate fracture criteria in the presence of voids to design structural or system components for ceramic materials [6–8]. Some experimental results have shown the geometric features of voids via scanning electron microscope. Farah used the field effect scanning electron microscopy (FESEM) to characterize the nano-alumina samples prepared by the sol–gel method. [9]. Mir studied the effects of different contents of nano-alumina ($Al_2O_3$) on strength and microstructural behavior, with each model characterized by scanning electron microscopy (SEM) and X-ray diffraction (XRD). According to the previous experimental observation results, the effects of void shapes in $Al_2O_3$ can be determined and used for multiple applications.

In addition to experimental works, molecular dynamics (MD) simulations have been widely used for studying deformation mechanisms of both ductile and brittle materials [10–12]. For example, Li et al. [13] studied the strengthening effects on modulus and softening effects on the fracture strength of nanoporous gold in multiaxial loading conditions by MD methods. Li et al. [14] focused on the deformation process of dislocation motion, grain boundary sliding phenomenon, and grain size rotation in nanocrystalline nanoporous metals in the nanoscale. In their study, Wei et al. [15] revealed the effects of grain size of nano-crystalline nickel titanium alloy based on the experimental observations of dynamic fracture processes under triaxial loading conditions in MD simulations. Qiu et al. [16] derived the effects of grain vacancy as well as the concentration rate on the nucleation of plasticity and spall strength in single-crystalline nickel material. Zhou et al. [17] studied the effects of randomly distributed voids on the damage evolution and fracture properties in single crystalline aluminum models. Sarker et al. [18] revealed the changes in elastic constants, energy decomposition under uniaxial loading, and triaxial tension conditions in atomic gypsum models and compared that with corresponding experimental results.

Altogether, the potential of combining atomistic models with larger scale models or experimental results can generate comprehensive insights into the mechanical and fracture properties under complex loading conditions, and this can inform the relationships between atomic structural deformation and material property changes [19–23].

In this paper, we highlight the importance of understanding the mechanical behavior of materials at the nanoscale under dynamic loading conditions. The use of MD methods and DFT in this paper enables the accurate calculation of the basic elastic constants of the material and the validation of the simulation results against experimental data. The study also shows that the fracture properties of the material are strongly influenced by the loading rates and directions, as well as the presence of voids. The results indicate that the fracture strain decreases more sensitively to strain rate than to void radius, which has important implications for the design and optimization of nanoporous material. Overall, the study provides valuable insights into the mechanical behavior of $\alpha$-$Al_2O_3$ under dynamic loading conditions, which can aid in the development of more robust and efficient ceramic materials for engineering applications.

## 2. Materials and Methods

To ensure the accuracy of the model, a precise atomic model consisting of lattice constants was initially selected and subsequently compared with reported results. Informed by our SEM (scanning electron microscope) and EDS (energy-dispersive X-ray spectroscopy) characterization works [24], the focus of this paper was to investigate the fracture behavior of $\alpha$-$Al_2O_3$. The atomistic model was constructed based on the structure of $\alpha$-$Al_2O_3$ utilizing six lattice parameters to define the unit cell. In the present study, the lattice parameters were obtained from Feng et al. [25] and are listed in Table 1. During the equilibration process, atom positions were adjusted under the canonical ensemble (NVT) at 300 K for 40 ps until the variation of length changes along different directions and total volumes were no greater than 5%.

**Table 1.** Lattice constant validation.

| Name | Unit Cell of Model | Theoretical Model [25] |
|:---:|:---:|:---:|
| *a* | 4.750 Å | 4.759 Å |
| *b* | 4.780 Å | 4.759 Å |
| *c* | 12.99 Å | 12.99 Å |
| *α* | 90° | 90° |
| *β* | 90° | 90° |
| *γ* | 120° | 120° |

To ensure the accuracy of the material structure, a validation process was essential prior to the actual loading. Firstly, a comparison between the extracted elastic constants (shear modulus, bulk modulus, and Poisson's ratio) in this study using the established $\alpha$-$Al_2O_3$ unit cell (refer to Table 1) and published works [26,27] is presented in Table 2. Elastic constants were obtained using both molecular dynamics (MD) and density functional theory (DFT) methods for better atomic structural and potential validation effects [28–30]. For MD, LAMMPS software was used for $\alpha$-$Al_2O_3$. Uniaxial tensile loading was applied separately in the X[100], Y[010], and Z[001] directions with periodic boundary conditions applied in all directions during the process. Uncertainty quantification was essential for elastic constant calculation [31–34]. The method based on the statistical nature of MD results was used here [35]. Firstly, models of different sizes with varying numbers of unit cells were constructed for the calculation of elastic constants. These sizes included $1 \times 1 \times 1, 2 \times 2 \times 2$, and $4 \times 4 \times 4$ unit cells. A total of 24 random systems were generated for each model size during a 50 ns equilibration process. To maintain the imposed strain, the NPT ensemble (300 K, 0 bar) was applied. After discarding the initial 10 ns, the stresses were time-averaged for the remaining 40 ns. The 40 ns time period was divided into 8 segments, and the bulk modulus, shear modulus, and Poisson's ratio were calculated for each segment. The final results were obtained by calculating the average value and standard deviation of the results from each segment. For DFT calculations, the PW91 (Perdew–Wang generalized-gradient approximation) and GGA (generalized gradient approximation) functions were used in CASTEP with the cut-off energy setting as 700 eV. The validation process in this study involved comparing the extracted elastic constants obtained from MD and DFT simulations with published experimental [26] and numerical values [27] . Usually, due to the computational cost and time required, MD simulations typically use strain rates of about 6 to 10 orders of magnitude larger than the highest strain rates commonly used in laboratory experiments. Based on these problems, quantitative analysis is an important way to analyze the relationship between micro-scale phenomena and macro-scale mechanical properties. This is achieved by comparing the dependence of the simulated bulk modulus and Poisson's ratio on the temperature and strain rate with the dependence observed in laboratory experiments performed by other researchers [36–38]. The calculated elastic constants were found to be within a reasonable range when compared with previous experimental and numerical results, with the maximum relative difference being 22% in shear modulus. This close agreement between the simulated and published results serves as a validation for the chosen structure and indicates its suitability for more complex loading conditions later in the paper.

**Table 2.** Elastic constant validation.

| Name | MD | DFT | Experimental Data [26] | DFT [27] |
|---|---|---|---|---|
| Shear modulus | 126 ± 5.6 GPa | 119 ± 6.4 GPa | 152 GPa | 158 GPa |
| Bulk modulus | 242 ± 8.6 GPa | 253 ± 10.3 GPa | 228 GPa | 247 GPa |
| Poisson's ratio | 0.23 ± 0.02 | 0.27 ± 0.02 | 0.22 | 0.24 |

In our models, the geometric features of atomistic alumina voids are determined via SEM [39,40] and TEM (transmission electron microscope) [41]. From the analysis of the void shapes, one basic void shape (i.e, the spherical voids) was chosen for multi-axial simulations. The void sizes are related to porosity, which are the ratio of void volume to total volume of the model.

The large-scale atomic/molecular massively parallel simulator (LAMMPS) was used to perform all simulations during the model construction and loading processes. Charge-optimized multi-body potentials (Comb3) developed by Choudhury et al. [42] were employed to describe the interactions between aluminum and oxide atoms of the alumina. Periodic boundary conditions were applied in the three orthogonal directions to avoid the influence of specimen dimensions, as illustrated in Figure 1. The pore structure in the

paper primarily consists of a single spherical void located at the 3D geometric center of the model with varying inner radii. The rationale for selecting the spherical shape as the void is based on the observation results from the SEM experimental findings [9,43,44]. By utilizing the most representative void shape, the impact of multiaxial loading on the trends and orientations of crack propagation can be more lucidly investigated under diverse stress states (e.g., strain rates). Other types of vacancies are excluded during the loading process to prevent potential interaction effects. In all simulation processes, the Velocity-Verlet and Leap-Frog algorithms are commonly used for numerical integration of motion, which differ mainly in velocity and position calculation requirements. The Velocity-Verlet algorithm is a superior choice for MD simulations due to its ability to prevent energy dissipation and its high computational efficiency compared with the Leap-Frog algorithm [45–49]. The initial specimen was relaxed to reach energy minimization states using the conjugate gradient method, with maximum force and energy tolerances of $10 \times 10^{-13}$ eV per square nanometer and $10 \times 10^{-15}$ eV per square nanometer, respectively. NPT ensemble is a popular accepted method for solid materials in molecular dynamics simulations due to its ability to adjust parameters to fit experimental environments [50–52]. Subsequently, the models were relaxed at 300 K and 0 bar for 80 ns under an isothermal–isobaric (NPT) ensemble in three directions until the potential energy and total volume values of the model fluctuated less than 5%, following the same standards of common published works [53,54]. Multi-axial loading was applied to all models using the stepwise straining method. The loading was applied to the simulation cell via the fix/deform procedure, with different strain rates ranging from $1 \times 10^9$/s to $1 \times 10^{11}$/s, and the specimen was remapped by coordinates throughout the loading process. The timestep length was 1 ps. All visualizations were obtained using Open Visualization software (OVITO) version 3.7.6, and the evolution of void structures was analyzed using the corresponding volumetric surface method [14]. To clarify the distinctions between nonporous and nanoporous models, we compiled the fundamental simulation parameters in Table 3. The table lists the number of atoms, loading directions, void dimensions, and other pertinent information.

**Table 3.** Model simulation parameter settings.

| Name | Atom Number | Strain Rate | Loading Orientation | Graphical Void Radius |
|---|---|---|---|---|
| Nonporous | 114,338 | $1 \times 10^9$–$1 \times 10^{11}$/s | Uniaxial, biaxial, triaxial | 0 Å |
| Nanoporous | 113,395 | $1 \times 10^9$–$1 \times 10^{11}$/s | triaxial | 12.5Å |
| | 111,042 | | | 19.0 Å |
| | 106,841 | | | 25.0 Å |
| | 98,615 | | | 32.0 Å |
| | 83,638 | | | 38.0 Å |

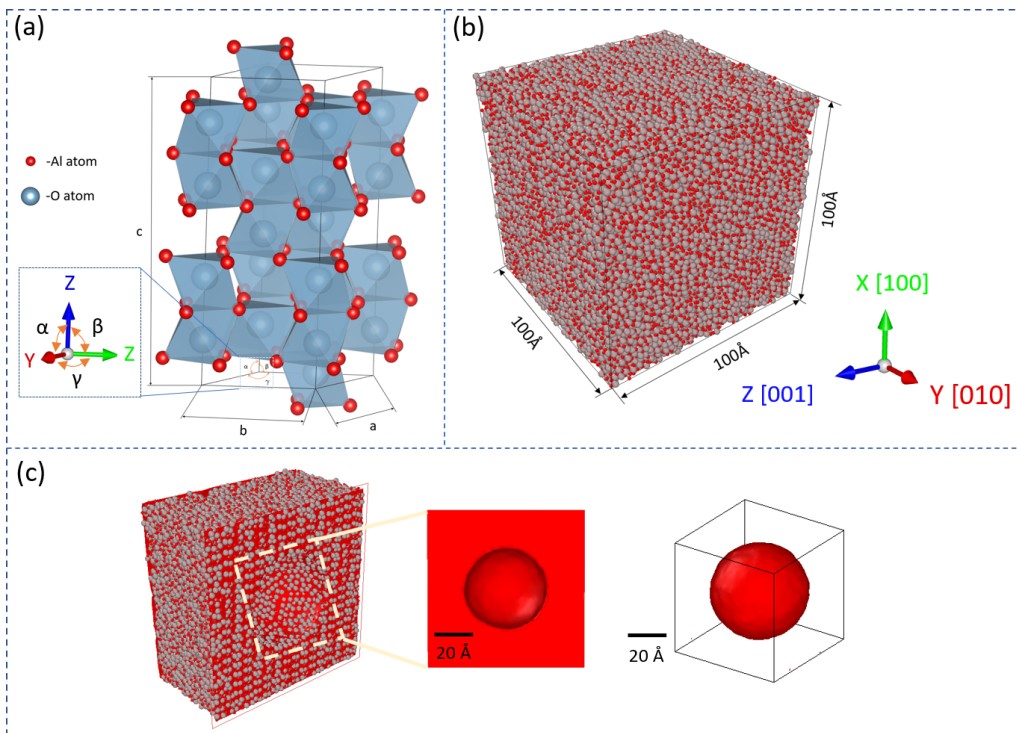

**Figure 1.** Atomic model introduction: (**a**) hexagonal unit cell of atomic $\alpha$-$Al_2O_3$ ceramic material. Atoms are colored according to their types, where red represents the aluminum atoms, and grey represents the oxygen atoms; (**b**) atomic configuration of $\alpha$-$Al_2O_3$ ceramics model; (**c**) a spherical void is positioned in the center.

## 3. Results

In this section, the validation results of the material models and interatomic potential file are presented using the validated data. The mechanical properties obtained from atomistic models under different loading conditions, such as multiaxial loading directions and strain rate, are also compared to investigate the changes in material response.

### 3.1. Effects of Stress State

From Figure 2a, it can be observed that the stress–strain curves of the nanoporous ceramics model under uniaxial tension were similar along the Y[010] and Z[001] directions, while the curve along the X[100] direction exhibited lower fracture stress and strain. The fracture strain along the Y[010] direction was the highest, followed by the Z[001] and X[100] directions. The fracture stress along the Y[010] and Z[001] directions was nearly the same, around 60 GPa, which was significantly higher than that along the X[100] direction. Additionally, the slopes of the curves were very similar in the elastic regime, indicating consistent elasticity of the model under uniaxial loading. Next, in Figure 2b, the stress–strain curves of nonporous $\alpha$-$Al_2O_3$ ceramics under biaxial tension loading are shown. The curves follow similar trends as the uniaxial case, but the values of fracture stress and strain along the Y[010] (54.5 GPa, 0.106) and Z[001] (50.3 GPa, 0.104) directions are larger compared with that along the X[100] direction. Finally, Figure 2c shows the stress–strain curves of nonporous $\alpha$-$Al_2O_3$ ceramics under triaxial tension loading. The values of fracture stress and strain along the Z[001] direction are the highest, and the values of slopes of the curves are very close for the X (40.4 GPa, 0.075), Y (43.3 GPa, 0.078), and Z (46.3 GPa, 0.079) directions, indicating decreasing effects on the stiffness compared with uniaxial loading.

Finally, a quantitative analysis was conducted on two crucial parameters, namely fracture strength and stiffness, of the nanoporous $\alpha$-$Al_2O_3$ ceramic model, under various loading conditions. The obtained results are presented in Figure 3, while the calculation

of the stiffness mainly relies on the slope of the elastic region from the corresponding stress–strain curve in Figure 2.

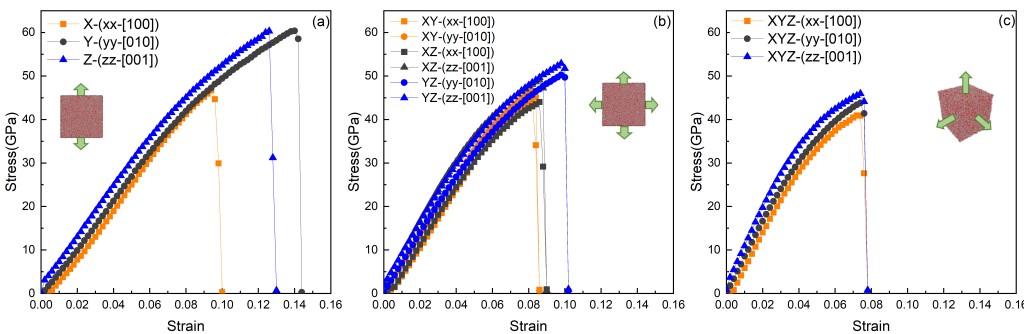

**Figure 2.** Stress–strain curves of nonporous $\alpha$-$Al_2O_3$ ceramics under (**a**) uniaxial tension, (**b**) biaxial tension, and (**c**) triaxial tension. For nomenclature, X-(xx-[010]), as an example, refers to the loading direction along the X direction (X) and stress responses from the direction (xx-[100]). The peak fracture stress and strain reduces from (**a**–**c**). The difference in peak fracture stress and strain becomes less pronounced under triaxial loading.

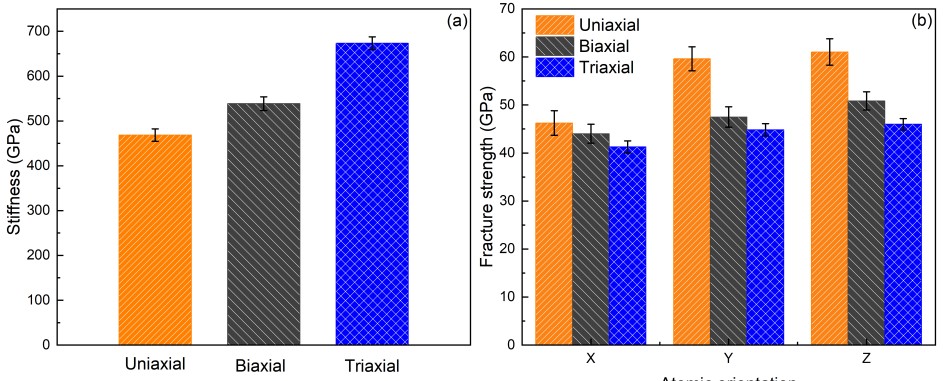

**Figure 3.** Mechanical properties of nonporous $\alpha$-$Al_2O_3$ models under uniaxial, bi-axial, and tri-axial tension at a strain rate of $1 \times 10^9$/s: (**a**) stiffness, (**b**) fracture strength. The Young's modulus increases as a function of triaxial loading. The fracture strength decreases under triaxial loading conditions.

For all models, the average stiffness and tensile strength under uniaxial tension was $457 \pm 11.6$ GPa and $47 \pm 5.2$ GPa, respectively. The current simulation-calculated stiffness closely matches the results published in existing literature (ranging from 401 to 688 GPa) [55,56], which verifies the accuracy of the models for further simulations. Since the atomic models possess an ideal unit cell structure and lack pre-existing atomic defects, the calculated stiffness values are marginally higher than the experimental results. Other differences could arise from the loading rate applied [13]. Nevertheless, as seen in Figure 3, the overall trends of both parameters are in line with the predicted increasing trends of stiffness, and this information could be useful in multi-scale modeling efforts [13]. From Figure 3, the results indicate that the stiffness under tri-axial loading conditions was the highest, with a maximum value of $662 \pm 14.3$ GPa, in contrast to the minimum value of $457 \pm 11.6$ GPa under uniaxial loading. The stiffness under biaxial loading was smaller ($538 \pm 12.5$ GPa) than that under triaxial loading but larger than that under uniaxial loading.

Concerning the ultimate fracture strength, the average under uniaxial loading was the largest ($55.3$ GPa $\pm 4.8$ GPa), while the average values under triaxial loading conditions were the smallest ($41.5$ GPa $\pm 4.2$ GPa). This phenomenon may be attributed to the inability of the atomic structures to produce broader plastic regions for deformation under the influence of mix-orientation loading, culminating in a diminished strength of the models.

### 3.2. Effects of Strain Rate on Crack Properties under Triaxial Loading Conditions

After comparing the effects of multiaxial and uniaxial loading conditions, we conducted a quantitative study on the influence of strain rates on the material properties of the nonporous $\alpha$-$Al_2O_3$ models. Previous studies on different materials have shown that the strain rate can affect material performance and cause structural changes under the intended loading conditions [13].

As shown in Figure 4a, the stress–strain responses, stiffness, fracture strength, and strain of the corresponding models were studied. The stress–strain curves along X[100], Y[010], and Z[001] directions exhibited similar increasing trends in the elastic stage. However, the trends of the curves were not entirely consistent with those in Figure 2 when the applied strain rate increased from $10 \times 10^9$/s to $10 \times 10^{11}$/s. The models subjected to the lower strain rate failed with the smallest strain values ($0.08 \pm 0.002$), while the models under the higher strain rate had higher failure strain values ($0.24 \pm 0.002$). Moreover, the curves under higher strain rates (blue) had higher peak stress and fracture strain compared to those of the curves (orange and grey) under lower strain rates.

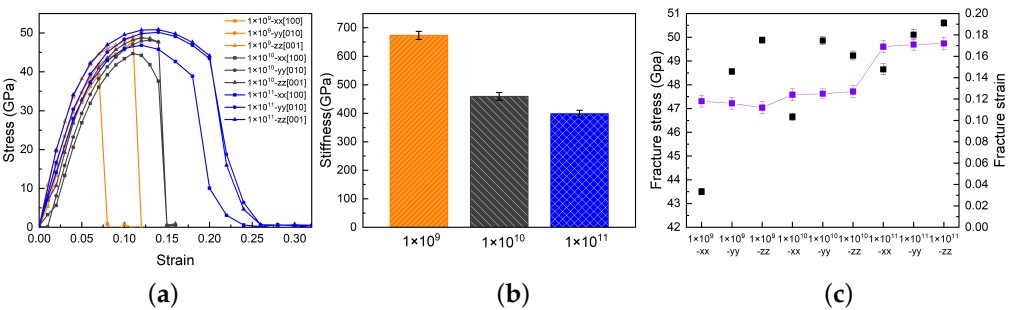

(a)                (b)                (c)

**Figure 4.** Mechanical properties of nonporous structures under uniaxial tension at different strain rates of $1 \times 10^9$/s, $1 \times 10^{10}$/s, and $1 \times 10^{11}$/s: (**a**) stress–strain curve, (**b**) stiffness, and (**c**) fracture strength and strain along X[100], Y[010], Z[001] directions. The peak fracture strain increases with increasing strain rates and the fracture stress increases from X[100] to Y[010] to Z[001].

The results show that the stiffness decreases from $662 \pm 14.3$ GPa to $310 \pm 11.7$ GPa in Figure 4b, but the ultimate fracture stress ($42.8 \pm 3.8$ GPa to $50.7 \pm 4.4$ GPa) and strain ($0.08 \pm 0.002$ to $0.24 \pm 0.002$) values increase with increasing strain rates in Figure 4c. Altogether, these results on the rate-dependent properties serve to feed higher-scale models [13] towards better predicting the mechanical response of alumina ceramics under impact and shock loading [57].

### 3.3. Effects of Porosity on Mechanical Properties and Fracture Surface

In this section, the deformation behaviors and microstructure evolution of nanoporous $\alpha$-$Al_2O_3$ models with a central void of different radii under triaxial tension are analyzed. The void volume fraction and deformation morphology of the atomic models are meticulously gathered and analyzed. Void volume fraction, defined as $\phi = N_\phi / N_0$, where $N_\phi$ and $N_0$ represent the volume of the void and the volume of the entire model, respectively. The stress–strain curves of the models are accompanied by captions that illustrate the crucial deformation morphology of the models, as follows.

The stress–strain curves along the X[100], Y[010], and Z[001] directions with void radii of 12.5 Å, 25 Å, and 38 Å are presented in Figure 5a. It can be observed from the curves that as the inner radius increases from 12.5 Å to 38 Å, the average ultimate fracture strength decreases in all directions (from 48.9 GPa to 26.3 GPa). Compared with the nonporous model, the decreasing effects on fracture strength are also noticed by comparison with that in Figure 4 (from 50.8 GPa to 26.3 GPa). Furthermore, larger fracture stresses (48.9 GPa vs. 26.3 GPa) are found along the Z [001] direction under decreased void radii (12.5 Å to 38 Å). The difference between the fracture stress along the Z [001] direction and that along the other directions is the smallest among all three void radii.

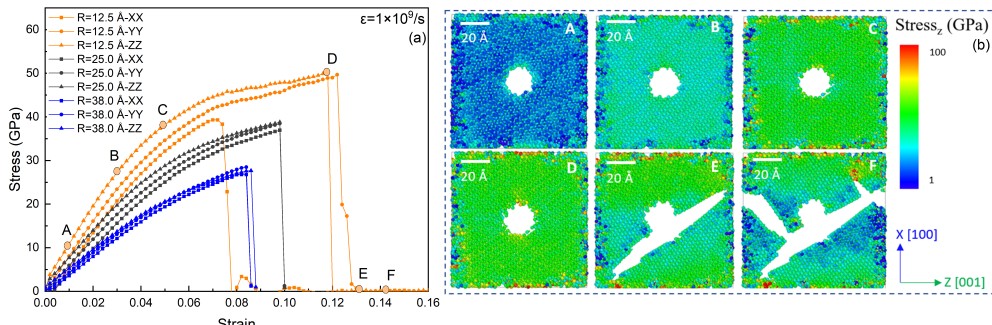

**Figure 5.** Stress –strain curves of nanoporous $\alpha$-$Al_2O_3$ ceramics under triaxial tension with an inner void radius R of 12.5 Å, 25 Å, and 38 Å are shown in (**a**). The configurations of model corresponding to the points on the stress–strain curve are illustrated on the right in (**b**). The peak failure stress and strain reduce from the (zz-[001]) direction to the (xx-[100]) direction. The difference in peak failure stress and strain becomes less prominent with the increase of void radius.

In addition, images of the deformation and fracture (on the right) have been mapped onto the figure at six critical strain points indicated in Figure 5a. Given the pivotal role of stress along the Z[001] direction in characterizing the material deformation morphology as established in the previous analysis, this direction has been selected. Notably, from the selected critical points A, B, and C, it can be observed that the average stress along the Z[001] direction increases almost linearly with a color transition from blue to green as per the color bar. However, at point D, stress concentration phenomena can be observed first around the spherical void. Due to the high brittleness of ceramic material, stress concentration mainly occurs in a small area of the model space compared to ductile materials [13]. Upon reaching the critical state (point E in the curve), the crack occurs at the edge of the sphere void, approximately at a 45-degree angle to the X[100] direction along the Y[010] direction, leading to the complete fracture of the entire model. Finally, point F represents the final stage of the model with a branch of the crack occurring, which illustrates the complete fracture state of the model.

The progression of the void volume fraction of the model with spherical voids of different sizes, namely 12.5 Å, 25 Å, and 38 Å, is depicted in Figure 6. The curve exhibits a quasi-linear increase in void volume fraction $\phi$ during the straining process, as illustrated in Figure 6a. However, it is important to note that the initial points of the models are different due to the varying size of the voids situated at the center of the models. The curves display similar increasing trends with the rise of applied strain until the models reach the critical fracture state, where the voids undergo significant expansion as indicated by the marked red points. The maximum $\phi$ value of 60.6% is observed in Figure 6b with an inner radius of 12.5 Å, while the minimum $\phi$ value of 51.8% was observed with a radius of 38.0 Å under the loading rate of $1 \times 10^{11}$/s, indicating that the models underwent complete fracture. During this process, cracks were observed at the center of the models, which serve as the primary source of the observed deformation results and contribute to the expansion of the void volume fraction. Furthermore, the branch of the crack shown in Figure 5a accelerates the fracture process of the models and also contributes to the sudden increase of the void volume fraction.

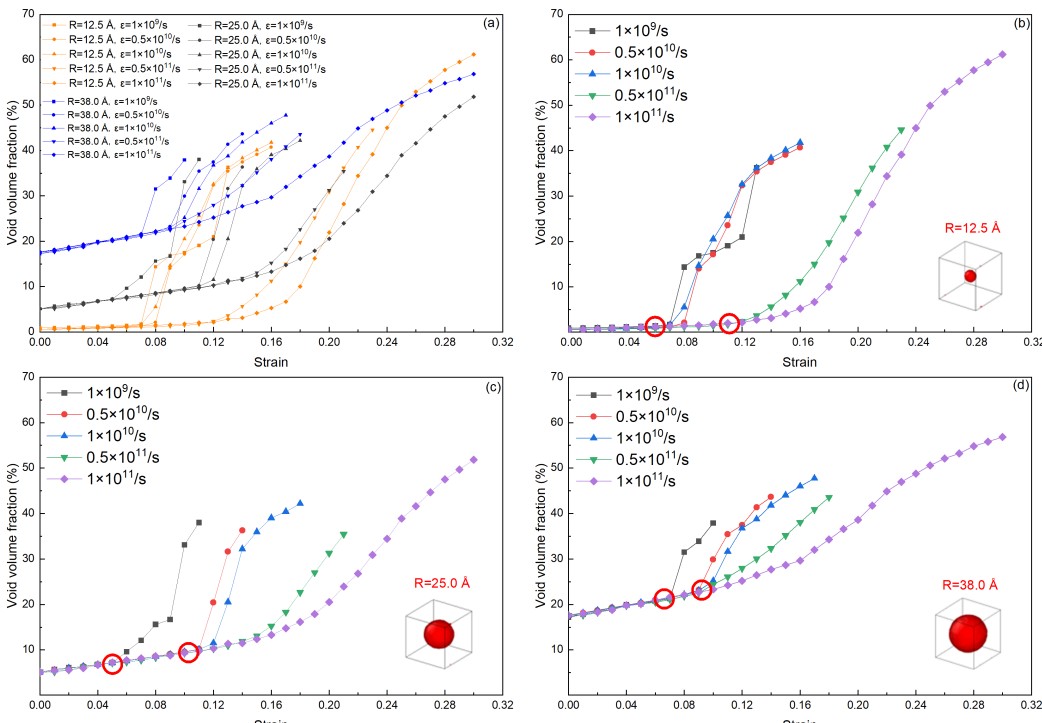

**Figure 6.** General void volume fraction $\phi$-strain curves of nanoporous $\alpha$-$Al_2O_3$ ceramics under triaxial tension with different inner void radius R ranging from $1 \times 10^9$/s to $1 \times 10^{11}$/s are shown in (**a**), and breakout curve for (**b**) R = 12.5 Å , (**c**) R = 25.0 Å, and (**d**) R = 38.0 Å. The increasing rate of the void volume fraction is similar across different initial radius. The marked red points denote the onset of rapid growth of the void.

Figure 7 displays the correlation between the average fracture strain and the radius of spherical voids under different loading rates ranging from $1 \times 10^9$/s to $1 \times 10^{11}$/s. In Figure 6, the key parameters are displayed to illustrate the changes in the ultimate void volume fraction of single model at different strain rates. In Figure 7, two additional simulation groups are depicted using the aforementioned parameters to illustrate the changing trends of the model's fracture strength. As the void radius increases from 12.5 Å to 38 Å, the fracture strain decreases gradually for different loading rates. Moreover, the fracture strain also increases with the increase of loading rate. When considering both parameters together, the maximum fracture strain of $0.24 \pm 0.002$ was observed when the void radius was 12.5 Å under the loading rate of $1 \times 10^{11}$/s, while the minimum fracture strain of $0.2 \pm 0.002$ occurred when the void radius was 38.0 Å. In contrast, the smallest fracture strain occurs when the loading rate was $1 \times 10^9$/s, with the maximum fracture strain of $0.118 \pm 0.002$ observed for R = 12.5 Å and the minimum strain of $(0.088 \pm 0.002)$ observed for R = 38.0 Å. Comparing the percentage decrease of fracture strain with the initial values 17% vs. 25% at R = 12.5 Å, it was concluded that the fracture strain was more sensitive to the strain rate than void radius.

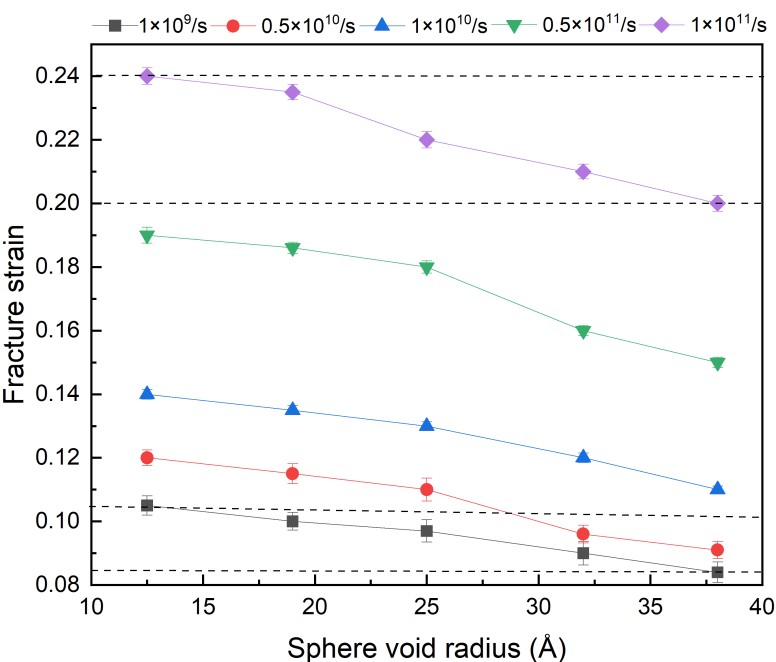

**Figure 7.** Fracture strain as a function of void radius R under triaxial tension loading with strain rates ranging from $1 \times 10^9$/s to $1 \times 10^{11}$/s. The fracture strain increases as a function of strain rate and decreases as a function of void radius, with decreases being more sensitive to strain rate. The dashed lines represent the maximum and minimum fracture strain values under corresponding strain rates and void radius.

## 4. Conclusions

In this work, the uniaxial, biaxial, and triaxial tension loading response of nanoporous $\alpha$-$Al_2O_3$ containing spherical voids of different sizes, and the deformation mechanisms and corresponding mechanical properties are compared and analyzed. First, the ceramic models are validated with both experimental and simulation results [26,27] (i.e., elastic constants, lattice constants). Next, the strengthening effects of stiffness and softening effects on fracture strength are investigated for multiaxial loading simulations and compared with the uniaxial loading condition. Compared with uniaxial loading, $\alpha$-$Al_2O_3$ ceramics are more affected by loadings in different directions, resulting in strengthening stiffness from $457 \pm 11.6$ to $662 \pm 14.3$ GPa, and ultimate strength $41.5 \pm 4.2$ GPa versus $55.3 \pm 4.8$ GPa. The effects of increasing strain rates lead to higher fracture strain and strength, and result in the decrease of stiffness values. Subsequently, the model deformation morphology depicts spherical voids of varying radii under triaxial loading conditions. An essential parameter, void volume fraction ($\phi$), was selected to describe the void volume evolution trends with structural deformation. The average stress along the Z[001] direction increases quasi-linearly, and the loading process highlights a stress concentration phenomenon at the edge of voids. The increased void radius across different models can lead to similar increasing trends of void volume fractions. The effects of increasing void radius on decreasing fracture strain are also depicted to be more sensitive under higher loading rates by comparing the percentage decrease of fracture strain (17% vs. 25%). These simulations can be used to investigate the connection between atomic microstructure deformation and macro-property changes when compared with high scale models and experiments.

**Author Contributions:** Conceptualization, J.C. and J.D.H.; methodology, J.C.; software, J.C.; validation, J.C. and J.D.H.; formal analysis, J.C.; investigation, J.C.; resources, J.D.H.; writing—original draft preparation, J.C.; writing—review and editing, Z.C. and J.D.H.; visualization, J.C.; supervision, J.D.H.; project administration, J.D.H.; funding acquisition, J.D.H. All authors have read and agreed to the published version of the manuscript.

**Funding:** This research was funded by China Scholarship Council (201806840127).

**Data Availability Statement:** Data will be made available on request.

**Acknowledgments:** This work is supported by Defence Research and Development Canada (DRDC), China Scholarship Council (201806840127). The views and conclusions contained in this document are those of the authors and should not be interpreted as representing the official policies, either expressed or implied, of General Dynamics, NP Aerospace, DRDC or the Government of Canada. The Government of Canada is authorized to reproduce and distribute reprints for Government purposes notwithstanding any copyright notation herein.

**Conflicts of Interest:** The authors declare no conflict of interest.

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
