# Peer review of "Molecular Dynamics Simulations Correlating Mechanical Property Changes of Alumina with Atomic Voids under Triaxial Tension Loading"

_2673-3951, doi:10.3390/modelling4020012_

Round 1

Reviewer 1 Report

The manuscript deals with the very difficult problem of simulating the mechanical properties and cracking of an Al2O3 material. The results obtained by the authors are certainly important for applications as well as knowledge about the dynamics of the appearance of a certain class of defects. The authors use both molecular dynamics and DFT methods where they use specialized software, e.g. LAMMPS.

(1) Authors wrote that ”In all simulation processes, the Velocity-Verlet algorithm was used for numerical integration of motion ...” Could the authors justify why? Why not at least a leap-frog algorithm? Then a comment on the (N,P,T) algorithm is required. How it is performed? What is the value of temperature?

(2) Authors wrote: ”During the equilibration process,

atom positions are adjusted under the desired temperature until the system reaches a stable state.” What do they understand by a stable state?

(3) The authors give the parameters of the unit cell. Could they include its scheme, the best together with Fig.1 or earlier?

(4) I do not understand the definition of the pore structure in the model.

Are they represent initially only the spherical voids of a given size? How they are distributed within the simulation sample. What about vacancies in the model, their motion, etc?

Minor:

Page 1 line 32 ”Li et al.[13]” and Page 2 line 34 ”Wang et al.[13]”?

The authors' results may be interesting for thin aluminum layers (sub-micrometer thick), where the mechanical properties of their outer layer of aluminum oxide may be important for the mechanical stability of the entire layer. Therefore, I suggest publishing this work after addressing the questions and comments.

Reviewer 2 Report

The authors studied the mechanical properties of nanoporous alumina through molecular dynamics simulations. The uniaxial, biaxial, and triaxial tension loading tests were performed and corresponding results were properly analyzed. The size effects of void on the mechanical properties were also examined. The paper is well organized and written. I thus recommend the paper for publication.

Minor comment:

·        Can the authors comment on the effect of the shape of the void? Do voids with other types of shapes behave similarly or differently as the spherical one?

Reviewer 3 Report

This manuscript performed molecular dynamics simulations to study the mechanical properties, e.g., stiffness, fracture train, fracture stress, etc., of nanoporous alumina under uniaxial, biaxial and triaxial tension loading conditions. This work firstly validated the MD model with available experimental and numerical results. Then, various deformation simulations were performed to investigate the effect of strain rate and void size. From my perspective, the following questions need to be addressed.

Minor comments:

  1. Table. 1 and Table. 2 have same captions

  2. Line 106: “The timestep length was 1ps, corresponding to an incremental strain of 0.001 per step.” But the authors previously mentioned the strain rate is 1e9 to 1e11.

  3. Section 3.1 & 3.2: “nonporous” and “nanoporous” are mixed up in the discussion. Such as line 127, 133, and Figure 2 and Figure 3. Please clarify which one is which. Is this part of discussion based on nonporous model systems?

  4. Figure 4: Add legend please.

  5. Figure 5: only Figure 5(a) is in the plot, where is Figure 5(b)? Please replot the figure and rephrase the corresponding discussion.

  6. Figure 6: the caption is not describing the correct order of subplots.

  7. Line 214: “while the minimum …” here might be “maximum”?

General comments:

  1. DFT is only used as validation and there are DFT results from other literatures as a baseline comparison. I don’t think the DFT is necessary in this manuscript as it is not being investigated in Discussion.

  2. Uncertainty analysis is needed for reliable results. Every loading condition needs multiple runs with statistically same model systems i.e., generated by the same parameters but different random states. And add standard deviation in all tables.

  3. An overview of experimental setup is needed. What model systems are generated and what loading conditions are performed, probably summarized in a table is good enough.

  4. Line 102: please add verification to show 80ps 300K 0 atm is long enough to make the system achieve equilibrium.

  5. Line 137 - 142: Firstly described as “closely matches” then “marginally higher” compared with experimental results. Is the comparison with experimental results based on the same strain rate? If not, please add more explanation about why the comparison is reliable.

  6. Figure 7: this figure contains void sizes that were not described in previous paragraphs. Please add an explanation for the void sizes in addition to 12.5, 25 and 38 Angstroms.

Round 2

Reviewer 1 Report

The corrections and clarifications made to the manuscript significantly improved its message. This version of the manuscript may be released for publication.

Author Response

Thank you so much for your patience and comments.

Reviewer 3 Report

Thanks for resolving my questions one by one, here are a few follow-up questions but should be minor:

  1. DFT validation: The DFT result is not used anywhere for investigation. I think it is not helpful to make the content more informative, but confusing instead. If the authors persist to keep it, please remove the highlight of DFT from the Abstract and Conclusion.

  2. Uncertainty quantification: thanks for adding this. But please add some detail of how those multiple systems are generated, especially how many systems do you have.

  3. Line 133: “the specimen was remapped following …” it was remapped by coordinates or velocity?

Author Response

Please see the attachment. Thank you for your help and patience.
